# Probiotics in the New Era of Human Milk Oligosaccharides (HMOs): HMO Utilization and Beneficial Effects of *Bifidobacterium longum* subsp. *infantis* M-63 on Infant Health

**DOI:** 10.3390/microorganisms12051014

**Published:** 2024-05-17

**Authors:** Chyn Boon Wong, Huidong Huang, Yibing Ning, Jinzhong Xiao

**Affiliations:** 1International Division, Morinaga Milk Industry Co., Ltd., 5-2, Higashi Shimbashi 1-Chome, Minato-ku, Tokyo 105-7122, Japan; 2Nutrition Research Institute, Junlebao Dairy Group Co., Ltd., 36 Shitong Road, Shijiazhuang 050221, China; 3Morinaga Milk Industry (Shanghai) Co., Ltd., Room 509 Longemont Yes Tower, No. 369 Kaixuan Road, Changning District, Shanghai 200050, China; 4Department of Microbiota Research, Graduate School of Medicine, Juntendo University, 2-1-1 Hongo, Bunkyo-ku, Tokyo 113-8421, Japan; 5Research Center for Probiotics, Department of Nutrition and Health, China Agricultural University, Beijing 100093, China

**Keywords:** probiotics, bifidobacteria, human milk oligosaccharides (HMOs), *Bifidobacterium longum* subsp. *infantis*, gut microbiota, infant health

## Abstract

A healthy gut microbiome is crucial for the immune system and overall development of infants. *Bifidobacterium* has been known to be a predominant species in the infant gut; however, an emerging concern is the apparent loss of this genus, in particular, *Bifidobacterium longum* subsp. *infantis* (*B. infantis*) in the gut microbiome of infants in industrialized nations, underscoring the importance of restoring this beneficial bacterium. With the growing understanding of the gut microbiome, probiotics, especially infant-type human-residential bifidobacteria (HRB) strains like *B. infantis*, are gaining prominence for their unique ability to utilize HMOs and positively influence infant health. This article delves into the physiology of a probiotic strain, *B. infantis* M-63, its symbiotic relationship with HMOs, and its potential in improving gastrointestinal and allergic conditions in infants and children. Moreover, this article critically assesses the role of HMOs and the emerging trend of supplementing infant formulas with the prebiotic HMOs, which serve as fuel for beneficial gut bacteria, thereby emulating the protective effects of breastfeeding. The review highlights the potential of combining *B. infantis* M-63 with HMOs as a feasible strategy to improve health outcomes in infants and children, acknowledging the complexities and requirements for further research in this area.

## 1. Introduction

In recent years, the role of the gut microbiome in early childhood development has garnered significant attention, leading to the progress of innovative approaches for the first 1000 days of life [1,2,3]. A cornerstone of this progress is the emergence of probiotics—live microorganisms that confer health benefits when administered in adequate amounts [4]—for infants. These beneficial bacteria, particularly *Bifidobacterium* strains, are increasingly recognized for their role in establishing a healthy intestinal microbiota during infancy, which in turn can influence overall health and development [5].

Human milk oligosaccharides (HMOs), complex sugars unique to human breast milk, have emerged as another critical component of infant nutrition [6,7]. These non-digestible carbohydrates are the third-most abundant solid component in human milk and serve as selective growth substrates for beneficial gut bacteria, particularly bifidobacteria [7]. The realization that these HMOs play a crucial role in shaping the infant gut microbiome has led to their incorporation into infant formula, attempting to mimic the composition and functional benefits of breast milk.

Given the intricate interplay between HMOs and the gut microbiota, attention is now turning towards the potential synergistic benefits of combining HMOs with probiotics, specifically those bifidobacterial strains capable of metabolizing HMOs. There is growing evidence to suggest that probiotics that can utilize HMOs, like *Bifidobacterium longum* subsp. *infantis* (*B. infantis*) strain M-63, exhibit unique physiological features that may amplify their clinical benefits [8]. M-63 has shown promise in promoting health throughout childhood, promoting healthy gut microbiota development, alleviating symptoms of gastrointestinal disorders, and reducing the incidence of allergic conditions.

The integration of HMOs and such specialized probiotics into infant formulas could provide a closer approximation to the benefits of breast milk, support the development of a healthy gut microbiome, and offer a proactive approach to promote child health. As we stand on the precipice of a new era in pediatric nutrition science, this review article aims to delve into the current state of knowledge surrounding the importance of bifidobacteria for infants, the rationale behind the incorporation of HMOs into infant formula, and the potential for reseeding infant’s gut with a particular strain of *B. infantis* M-63 to promote their health throughout childhood. Through a critical exploration, we will discuss the proven clinical benefits of the specific probiotic strain *B. infantis* M-63 and consider the future implications of these advances on infant and children’s health.

## 2. Importance of Bifidobacteria in Early Life

The establishment of a healthy gut microbiota during the initial stages of life profoundly impacts an individual’s later health. Within the microbial community acquired during early life, the genus *Bifidobacterium* holds particular significance, exerting essential functions in the health and development of newborns and infants [5]. Bifidobacteria are Gram-positive, non-spore-forming, non-motile, and non-filamentous strict anaerobic bacteria. The early dominance of bifidobacteria is significant due to their beneficial properties. Their pivotal role spans various aspects of gastrointestinal health, immune system maturation, and overall well-being, making them indispensable players in the complex ecosystem of the infant gut microbiota [9].

Ongoing research into bifidobacteria has unveiled their intricate interactions within the gut ecosystem, involving both microbe–microbe and host–microbe dynamics [10,11]. These interactions have significant biological implications, including the cooperative breakdown of complex, indigestible carbohydrates, which leads to the generation of short-chain fatty acids (SCFAs) like acetate and butyrate. In the early stages of life, acetate from bifidobacteria is essential for reinforcing the integrity of the intestinal barrier and supporting the maturation of the immune system, while also providing vital nutrients to intestinal cells and engaging in cross-feeding interactions that benefit the overall gut microbiota ecosystem [12]. Cross-feeding is a particularly advantageous strategy employed by bifidobacteria [13], where bifidobacteria act as primary degraders, metabolizing carbohydrates and resulting in the production of SCFAs such as acetate and lactate, which are subsequently utilized by secondary degraders, including butyrate-producing bacteria. For instance, although bifidobacteria are incapable of producing butyrate autonomously, they establish a mutualistic relationship with butyrate-producing bacteria such as *Faecalibacterium prausnitzii*, where the acetate that is produced by bifidobacterial species is used as a precursor for butyrate synthesis [14]. This cooperative interaction facilitates their mutual survival and function within the gut by sharing metabolic by-products and enhances the fitness of the entire gut microbiota.

Beyond their role in fermenting indigestible glycans, bifidobacteria also interact with the host’s immune system, playing a pivotal role in stimulating and modulating both innate and adaptive immune responses. Bifidobacteria were shown to potentially enhance the effectiveness of vaccines through improvements in immunologic memory [15,16]. A prospective observational study investigated the connection between *Bifidobacterium* levels in the feces of healthy infants around the time of vaccination and their subsequent T-cell and antibody responses [15]. The infants underwent vaccination procedures that included Bacillus Calmette–Guérin (BCG), oral polio virus, tetanus toxoid (TT), and hepatitis B virus, with *Bifidobacterium* levels measured at key intervals. The study found that greater early-life abundance of *Bifidobacterium* was linked with stronger CD4 T-cell responses to BCG, TT, and hepatitis B virus, as well as with higher plasma TT-specific immunoglobulin G (IgG) and stool polio-specific IgA after two years. Notably, these correlations were also prominent for the main subspecies encountered, *B. infantis* [15]. This evidence underscores the significance of *Bifidobacterium*, particularly *B*. *infantis*, in shaping early immune function and vaccine responsiveness.

Moreover, bifidobacteria have developed sophisticated genetic responses to confront bile, a critical factor in their survival within the gut environment due to the antimicrobial effects of bile. The presence of bile salt hydrolases (BSHs) enables bifidobacteria to neutralize bile salts, facilitating not just their own persistence but also leading to the production of by-products with beneficial traits, such as cholesterol reduction [17]. This presence of BSH activity, particularly in strains isolated from primates, is telling of the deep evolutionary history shared between these bacteria and their mammalian hosts [18]. Altogether, the current understanding on the role of bifidobacteria in early life has prompted efforts to supplement infant formulas with specific strains of bifidobacteria, recognizing their role in promoting correct microbial and immune system development and preventing non-communicable diseases.

## 3. Ecology of Bifidobacteria

The *Bifidobacterium* genus currently consists of 101 (sub)species, with the number steadily increasing over the past years [19]. Bifidobacteria species have been found not only in human guts but also in the digestive systems of various mammalian animals and environments such as sewage, the vaginal cavity, the mouth, breast milk, and certain types of food [20]. Particularly in social species—like mammals, birds, and insects, which rely on nurturing from parents—the connection of bifidobacteria from mother to offspring is typical, indicating an evolutionary bond that closely links parents with their young and the associated microbial communities [21]. Based on genomic data, the *Bifidobacterium* genus can be divided into seven main phylogenetic/phylogenomic clusters that are named after one of the species, and they include the following: *Bifidobacterium bifidum* cluster, *Bifidobacterium longum* cluster, *Bifidobacterium pseudolongum* cluster, *Bifidobacterium adolescentis* cluster, *Bifidobacterium pullorum* cluster, *Bifidobacterium asteroides* cluster, and *Bifidobacterium boum* cluster [22]. These clusters show genetic variability and phenotypic/physiological differences, indicating genetic adaptations to different ecological niches within the mammalian gastrointestinal tract. The close relationship between bifidobacteria and their hosts, particularly in social species that rely on parental care, suggests a microbe–host co-evolution scenario.

In the human host, *Bifidobacterium* species that naturally occur in the human gastrointestinal tract are referred to as the human-residential bifidoabcteria (HRB) (Figure 1) [23,24]. Within the HRB group, bifidobacterial species that are prevalent in the infant gut, including *Bifidobacterium longum* subsp. *infantis*, *B. breve*, *B. bifidum*, and *B. longum* subsp. *longum*, are sub-categorized as infant-type HRB [25]. Conversely, other bifidobacterial species that are more typically found in the intestines of adults, including *B. longum* subsp. *longum*, *B. adolescentis*, *B. catenulatum*, *B. pseudocatenulatum*, etc., are grouped as adult-type HRB [26]. Nevertheless, the classification into infant or adult types is not strictly defined, as bifidobacterial (sub)species do not apparently separate along these age-based lines [25,27]. Some HRB strains appear equivalently adapted to various human habitats—such as the gut, bloodstream, or breast milk—and different stages of life, from infancy to adulthood, with no clear phylogenetic or genomic distinctions observed for the studied strains [28,29]. Specifically, *B. longum* subsp. *longum* is remarkable for its presence in both infants and adults [30]. The widespread occurrence of this species throughout different stages of human life implies a genetic variability that likely improves its adaptive capabilities in the human gut environment. This extensive distribution is in part attributed to broad transmission within families, which extends beyond the mother–infant relationship and includes other familial connections [30,31].

In contrast, bifidobacterial species that are commonly found in animals or environmental settings, including *B. animalis* subsp. *animalis*, *B. animalis* subsp. *lactis*, *B. thermophilum*, *B. pseudologum*, etc., are referred to as non-HRB species (Figure 1). These non-HRB species are bifidobacterial species that have become specialized to inhabit specific animal digestive tracts [32]. Notably, *B. animalis* subsp. *lactis*, generally originating from animal gut and frequently used as a probiotic in dairy products, is occasionally identified within human feces [33]. This occurrence may be linked to dietary habits, like yogurt consumption [34], implying that this subspecies is not typically a commensal within the human gut (non-HRB) [35].

Accumulating evidence shows that the HRB species demonstrate a profound co-evolutionary relationship with their human hosts. A hallmark of this adaptation is the HRB species’ saccharolytic capability—their metabolism is deeply intertwined with the breakdown of complex glycans, including those that are host-derived, like HMOs and mucin, and diet-derived plant carbohydrates, indicating a mutualistic bond with the human host [36,37]. The genomes of HRB species are rich in the genetic apparatus necessary for the utilization of complex carbohydrates that are inedible to their human hosts [36,37]. This ability is not just a marker of their ecological niche within the gut but also influences the larger makeup of the gut microbiome.

Specific to infancy, infant-type HRB species display a remarkable capability to utilize HMOs, thanks to their suite of glycosyl hydrolases (GHs). Notable species like *B. bifidum*, *B. infantis*, and *B. breve* dominate the gut microbiota of breast-fed infants because of their unique proficiency at transforming these otherwise indigestible substances—an evolutionary trait likely driven by the critical period of neonatal development [38]. In addition to the HMO utilization capability, the order in which species arrive in the infant gut can also influence which *Bifidobacterium* species become predominant [39]. In a recent study that examined how different *Bifidobacterium* species commonly found in the infant gut establish dominance and impact the assembly of the gut microbiota, *B. breve*, which is noted as a weak competitor with limited HMO-utilization ability, can still become predominant if it colonizes early. The study found that regardless of its HMO utilization ability, *B. breve* can still outcompete the other stronger competitors if it arrives earlier, highlighting the impact of the priority effect on early-life gut microbial assembly [39]. Collectively, the presence of infant-type HRB species reflects a specialized adaptation to the infant gut, signifying a co-evolution that offers nutritional and developmental benefits during the earliest stages of human life.

Altogether, these findings demonstrate a symbiotic relationship between the HRB species and the human hosts, with the HRB species exhibiting traits especially well-suited for the human gut. The HRB’s metabolic activities closely align with the host’s dietary intake, evolving to exploit HMOs in infancy and diverse polysaccharides in adulthood [38]. This deep-seated co-evolution signifies that the presence of HRB in the gut microbiota is essential for human health, playing significant roles from early development and continuing to impact the gut ecosystem’s health and function throughout life [40]. This long-term co-evolution has led to a mutualistic relationship where HRB and their human host experience reciprocal benefits from one another’s existence and biological functions.

## 4. Infants Are Losing a Key *Bifidobacterium* Species

As outlined above, establishment of a bifidobacteria-dominant gut microbiota, which is recognized as a hallmark of normal infant gut microbiota, is crucial for healthy gastrointestinal and immune development. Delayed colonization of bifidobacteria, particularly infant-type HRB species, can negatively impact the health status of an infant and their subsequent development [9]. A number of factors contribute to the aberrant colonization of bifidobacteria, including disruptions to vertical transmission (for example, whether there was a vaginal or caesarean-section birth and intrapartum antibiotic use) [41,42] and disruptions to horizontal transmission (notably, feeding methods such as breastmilk or formula) [42,43]. The mode of delivery plays a defining role in the initial microbial exposure, subsequently shaping colonization of the infant gut. Vaginal delivery typically results in the transfer of beneficial maternal microbiota, including bifidobacteria, to the newborn. In contrast, cesarean section can lead to a delay in microbial colonization and a different composition of the infant’s gut microbiome [44]. Similarly, the use of antibiotics can disrupt the delicate balance of developing gut microbiota, and formula feeding may not provide the unique oligosaccharides found in human breast milk that selectively promote the growth of beneficial microbes like infant-type HRB [45,46].

### 4.1. The Declining Prevalence of B. infantis

Traditionally, the infant-type HRB, including *B. breve*, *B. longum*, *B. infantis*, are commonly found in the gut of breastfed infants due to their competitive advantages in utilizing HMOs from breast milk [25]. In particular, *B. infantis* uniquely contain a 43 kb gene cluster that encodes glycosidases and oligosaccharide transport proteins, allowing it to utilize and outgrow the other bifidobacterial species in the presence of HMOs [47]. However, emerging evidence points to the occurrence of microbiota change in infants, signified by the low detection rate of *B. infantis* in industrialized countries. The changes in lifestyle, including reduced breastfeeding, increased antibiotic use, and an increased number of cesarean sections, are speculated to be potential factors causing the lower prevalence of *B. infantis* [47,48].

Reduced breastfeeding remarkably decreases HMO content, the preferred source of nutrition for *B. infantis*, from the infant diet. It also reduces the chances of horizontal transmission of *B. infantis* among breastfed infants. All infant formulas from the past century do not contain HMOs comparable to human breast milk due to the technical limitations in large-scale production of HMOs until the recent decade. *B. infantis* are essentially depleted of its preferred source of nutrition in formula-fed infants. Therefore, it is suggested that the convenient switch from breastfeeding to formula feeding has removed the typical microbiota-accessible carbohydrates (MAC) from infants, causing the progressive and irreversible disappearance of *B. infantis*, as well as the overall change in the microbiota of infant populations over the past generations [43]. In cohort studies, a lower abundance of *Bifidobacterium*, specifically *B. infantis*, as well as higher microbial diversity are generally observed in formula-fed infants as compared to breastfed infants [49,50]. The growing effect of the increasing popularity of formula diet replacement reduces the generational succession of *B. infantis*. Regardless of the feeding choices, it is shown that the prevalence of *B. infantis* in infants is dropping [51,52]. In extreme cases, it has also been reported that *B. infantis* is never detected in the first 6 months of life in a portion of infants [53].

A recent study also pointed out that the ongoing persistence of *B. infantis* in populations may depend on the horizontal transmission between breastfed infants, as this species may not transmit vertically from the mother, because it was observed to be present as a part of the infants’ gut microbiome at 2 months of age [43]. As such, if breastfeeding is interrupted, the species could be at risk since it needs contact between breastfed infants for persistence. Evidence suggests that in Western societies, where breastfeeding and contact opportunities between breastfed infants have declined, this infant-type HRB species is now rare, even in babies not given antibiotics [54]. The species-specific loss of *B. infantis* is also thought to be because this species is not a typical colonizer of the adult gut and partially due to its eradication from the maternal gut microbiome as a result of maternal antibiotic use and other practices that significantly impact the microbiome [54]. Consequently, if the mother’s gut lacks *B. infantis*, the infant misses the chance to receive this crucial bacterium during birth, which is the primary means by which newborns establish their own gut microbiome.

These studies have clearly pointed out that *B*. *infantis*—a pivotal subspecies of *Bifidobacterium* that can efficiently utilize HMOs—faces the risk of becoming extinct [54,55], with its apparent loss in parts of the world with historically lower breastfeeding rates and shorter breastfeeding durations [56].

### 4.2. The Role of B. infantis in Early-Life Development

*B. infantis* is an important colonizer of the neonatal gut, with multiple physiological functions including modulating the immune system, influencing the overall nutrition status of infants and most recently, promoting bone formation and brain injury recovery [57,58,59,60].

It is proposed that *B. infantis* could modulate the infant’s immune response by accelerating the maturation of immune response, balancing the immune system to suppress inflammation, improving intestinal barrier function and increasing acid production [8]. *B. infantis* is considered a critical strain in the infant gut for the establishment of colonization resistance, in which the pH of the gut is lowered by acid production to reduce the growth of opportunistic pathogens [61,62]. Current randomized control trials have consistently proven that the supplementation of *B. infantis* will significantly lower the fecal pH to around 5.1, while remaining at the pathogen-favoring level of 6.0 in the control group [63,64,65]. In the case of delayed *B. infantis* colonization, the levels of acetate and lactate are also remarkably reduced, leading to a greater colonic pH that does not provide colonization resistance to harmful pathogens. Research from cohort studies reveals that the absence of this protective mechanism and the consequent raised pH levels in the guts of infants who are breastfed are linked to greater amounts of potentially harmful pathogens, increased antibiotic-resistance gene load, and chronic intestinal inflammation in the initial 100 days post birth, which are closely related to a higher occurrence of autoimmune diseases [66].

In addition, observational studies indicated that the reduced abundance of *B. infantis* is associated with severe acute malnutrition (SAM) in Bangladeshi infants, suggesting a role of *B. infantis* in impacting the neonatal nutrition status [67]. For healthy Bangladeshi infants, *B. infantis* represents ~75% of all bifidobacterial strains detected in fecal samples throughout the first year of age due to its rapid colonization in the first month. Infants with SAM, on the other hand, showed a relatively lower count (2–3 orders of magnitude) of *B. infantis*. Instead, potential pathogens, including *Escherichia*, *Shigella*, *Klebsiella*, and *Streptococcus* species, were found to be the dominating species. The positive association between malnutrition in children and impaired gut microbiota development is also supported in other studies, with speculated causes being dysbiosis-induced diarrhea and environmental exposure to pathogens, etc. [68,69,70]. Mediating the disrupted microbiota through the supplementation of a suitable probiotics or a synbiotic, as a result, has been proposed as a potential clinical approach to improve nutrition and health [71]. A subsequent clinical trial on the Bangladeshi SAM infants revealed that the supplementation of *B. infantis*, with or without lacto-N-neotetraose (LNnT), significantly improved the abundance of *B. infantis* and increased the weight gain of infants as compared to the placebo group. Nevertheless, despite the supplementation, the abundance of *B. infantis* remained to be 1–2 orders lower than in the healthy infants, potentially attributed to the low breastfeeding rate in these SAM infants. Nonetheless, the partial restoration of gut microbiota and improvement of growth in SAM infants adds to the growing evidence that supports the importance of *B. infantis* for infant and its potential clinical application in combating the global threat of undernutrition in children.

Emerging preclinical studies suggest that *B. infantis* are important for the skeletal system through the gut–bone axis and the neural system through the gut–brain axis [58,59]. Supplementation of *B. infantis* positively impacted the overall bone length and the activity of bone-remodeling cells by stimulating the PI3K/AKT pathway through the GH/IGF-1 axis in growing mice. Similarly, the supplementation of *B. infantis* is also shown to modulate the neuroinflammatory response and subsequently the antibiotic-induced hypoxia-ischemia brain injury by lowering the pro-inflammatory cytokines in neonatal mice.

## 5. The New HMO Eras in Infant Nutrition

Human milk oligosaccharides (HMOs) are important bioactive compounds that represent the third most abundant solid components in human milk, with over 200 structurally different glycans identified [7]. HMOs are comprised of five monosaccharides linked by glycosidic bonds, including glucose, galactose, fucose, N-acetylglucosamine, and sialic acid. These indigestible carbohydrates vary significantly in both concentration and types among mothers. Colostrum is typically found to have higher concentration of HMO at ~20 g/L, which drops to ~13 g/L in mature milk. Distinctively different HMO profiles are observed in different human milk samples due to a number of factors, with the most prominent being the gene-determined expression of glycosyltransferases by the mother [72,73]. About 70% of the mothers carry the secretor gene that encodes the enzyme α-1-2-fucosyltransferase (FUT), which favors the synthesis of α-1-2-fucosylated HMO like 2′-fucosyllactose (2′-FL) [74].

HMOs are expected to become next-generation innovative ingredients in the infant nutrition industry, not only for the fact that the addition of HMOs bridges the compositional gap between infant formula and human milk, but also due to the ability of HMOs to deliver certain health benefits similar to breastfeeding. Before the commercialization of HMO in recent years, short- and long-chain dietary fibers, including galacto-oligosaccharides (GOS) and fructo-oligosaccharides (FOS), were commonly added to infant formulas to mimic the functions of HMOs in the neonatal gut [75,76,77,78]. Nowadays, major HMO components from breast milk, including 2′-FL, 3′-sialyllactose (3′-SL), 6′-sialyllactose (6′-SL), lacto-N-neotetraose (LNnT), and a growing number of HMO glycans are becoming commercially available. The safety of these HMOs is supported in clinical trials, with the addition level ranging from 0.2 g/L to 5.75 g/L. The efficacy of HMO supplementation in modulating gut microbiota composition and mediating the immune response have also been studied [79,80,81,82,83]. Collectively, these human clinical studies propel the regulatory approvals for the use of HMOs in infant formula in different countries.

### 5.1. Physiological Functions of HMOs

The primary function of HMOs for infants is to be the ‘food for bugs’. Although a minimal percentage of HMOs are absorbed and enter circulation, the majority of HMOs are able to resist the gastric acidity and the hydrolysis by host enzymes, eventually reaching the intestine intact [7]. In the gut, HMOs exert biological functions through direct interactions with epithelial cells and pathogens, or indirectly through metabolites picked up by the various targets in and beyond the gut [84]. Decades of research has revealed the many roles of HMOs in the infant gut, including serving as the mediator of the gut microbiota, the anti-adhesive to pathogens, the stimulator of immune responses, and the modulator of intestinal epithelial cells.

HMOs mediate the gut microbiota and contribute significantly to the colonization of *B. infantis* during the early infancy. A consortium of bifidobacteria is often found in the newborn gut, while *B. infantis* is not present until a few weeks later. It is suggested that without the presence of HMOs, *B. infantis* would not be able to ‘overturn’ the established dominance of bifidobacterial species and claim its own [53]. Apart from *B. infantis*, HMOs have already been shown to be effective in positively shaping the microbiota of infant gut, which is especially important for the formula-fed infants. The gut microbiota of formula-fed infants are often characterized by a higher degree of microbial diversity, lower abundance of bifidobacteria and higher count of pathogens than breastfed infants, all of which are shown to be improved to some extent by the supplementation of different HMO mixtures in the formula, as substantiated in recent clinical studies [81,85,86].

Structural similarity between the HMO glycans and epithelial cell surface glycocalyx allows for the binding of pathogens to HMO molecules instead of the epithelia cell, potentially preventing the invasion of pathogens to the host and resulting in reduced incidence of infections [84]. In vitro and animal model studies have demonstrated that HMOs can act as receptor decoys to prevent the attachment of viruses like enterovirus, caliciviridae, and norovirus, bacteria like *Campylobacter jejuni*, and parasites like *Entamoeba histolytica* [87,88,89,90]. Recent studies also indicate that the antimicrobial effect of HMOs extend beyond merely the prevention of pathogen attachment to host cells. It is shown that HMOs could significantly inhibit the formation of biofilms by *Staphylococcus aureus*, *Streptococcus agalactiae* and other biofilm-producing bacteria [91]. Furthermore, HMOs have also been shown to be able to directly alter the morphogenesis of microbes, which subsequently inhibit pathogen attachment to host cells [92]. To date, however, human intervention studies are lacking information to confirm the exact dosage and combination of HMOs required for delivering the desired anti-adhesive effects.

HMOs, especially sialyllactoses, can also directly affect the proliferation, differentiation and apoptosis of epithelial cells in the gut and in other parts of the body like the urinary tract, potentially rendering enhanced protection to host cells against pathogen invasion [93,94]. HMO glycans are also shown to bind to immune cell receptors and hence modulate immune responses. For example, the release of relevant cytokines by macrophages and dendritic cells are reported to be influenced by HMOs in cell line studies [95,96,97]. In human studies, immunity enhancing effects are also studied in infant population of different health statuses. Notably, the supplementation of HMOs in infant formula was found to result in lower plasma inflammatory cytokines in healthy infants, reduced morbidity, medication use, and antibiotic use in allergic infants [86,98].

### 5.2. Maximizing the Effect of HMO Supplementation with B. infantis

Based on the average concentration in breast milk samples, different HMO mixes are designed, studied, and eventually applied in infant formulas. However, the benefit of HMO supplementation in formula may not be fully realized in the absence of *B. infantis*. It has been shown in a clinical trial that HMOs from breastmilk are not fully utilized and are largely eliminated in the stool from the one-month-old infants in the control group (no *B. infantis* supplementation), while the infants receiving supplementation of *B. infantis* showed a 10-fold reduction in fecal HMOs [63]. Although other infant-type HRB species, including *B. breve*, *B. longum*, and *B. bifidum*, have also been shown to utilize HMO, they are genetically not as efficient and capable as *B. infantis* and thus are unable to maximize the use of HMOs [99]. Indirect evidence was reported from a Danish cohort, where the rapid growth of *B. infantis* in the first few months of life matched with the rapid reduction in fucosyllactoses and sialyllactoses tested in the fecal samples from breastfed infants but not for the other infant-type HRB species [53].

These observations aligned with the presumed synergy between HMOs and *B. infantis* and support the idea that the maximized HMO utilization depends on the presence of *B. infantis*. Mechanistically, a carefully designed mice study came to the conclusion that the recovery of lost microbiota species over generations of changed diet is not feasible with just the administration of the original diet, but requires the administration of both the missing taxa and the original diet at the same time [100]. Analogizing the same theory in the context of infants, the lost microbiota species would be the *B. infantis* as stated above, while the changed diet would be a change from breastfeeding to the HMO-lacking infant formula. Therefore, in order to maximize the effect of HMO supplementation in formula, simultaneous supplementation of the probiotics strain of *B. infantis* would be helpful to ensure not only the immediate benefit of improving HMO utilization, but also the long-term benefit of restoring the endangered key infant microbial species.

## 6. *B. infantis* M-63: An Efficient HMO Utilizer

Supplementation with *B. infantis* can be considered an effective strategy to mitigate the impact of delayed colonization or the apparent loss of this particular subspecies and emulate the health benefits observed in vaginally delivered and breastfed infants. In this context, *B. infantis* M-63, which is a well-established, clinically studied infant-type HRB probiotic strain, is particularly adept at thriving in this environment due to its extensive genetic repertoire for HMO metabolism [101]. Several studies have documented the positive effects of *B. infantis* M-63 supplementation as a single strain or in combination with other probiotic strains on infant and children health, particularly in supporting premature infants [102,103], promoting bifidobacteria colonization [65,104,105,106], improving gastrointestinal conditions [107,108,109], and alleviating allergies [110,111]. This strain has demonstrated the capability to not only efficiently utilize HMOs [112,113,114] but also to promote healthy gut and immune system development via metabolite production [115,116,117], which is critical for overall health throughout childhood. Supplementation with this probiotic strain has the potential to offer a sort of “reseeding” effect in situations where the natural colonization process has been compromised.

### 6.1. Identification of B. infantis M-63

*B. infantis* M-63 (designated as M-63) has a well-documented safety, clinical efficacy, and application profile, supporting its status as a safe probiotic strain for use in a range of food products aimed at infant nutrition and beyond. Isolated in 1963 from the feces of a healthy infant, M-63 was identified as *B. longum* subsp. *infantis* based on comprehensive morphological, physiological, and genetic characteristics. M-63 is characterized by its rod-shaped, Gram-positive, anaerobic, non-motile, and non-spore-forming attributes, with strong adherence activity to human gastrointestinal tract [118]. Over the years, M-63 has been incorporated into a wide range of commercially available products such as infant milk formula, young child formula, and dietary supplements in the forms of sachet powder, oil drop, capsule, and tablet. Its lyophilized powder—manufactured by Morinaga Milk Industry Co., Ltd. (Tokyo, Japan)—is renowned for its excellent stability during storage and high survivability in finished products, such as powdered formula, until consumption [119].

### 6.2. Safety of B. infantis M-63

M-63 is well evaluated for safety whereby a comprehensive safety evaluation of M-63, which includes functional, genomic, and in vivo analyses, demonstrated that M-63 is a non-pathogenic, non-toxigenic, non-hemolytic, and non-antibiotic-resistant probiotic bacterium that does not contain any plasmids and does not display harmful metabolic activities [118,120,121,122].

The prominent safety profile of M-63 is also substantiated by in vitro, in vivo, and human clinical studies published in peer-reviewed scientific journals. It is noted that M-63 produces L-lactic acid without generating D-lactic acid [120], a form that can accumulate and cause metabolic complications in humans, especially in those with short bowel syndrome. The ability of M-63 to produce L-lactic acid is a positive indicator of its safety for human consumption. Importantly, M-63 also displays conjugated bile salt hydrolytic activity, meaning it can hydrolyze conjugated bile acids such as taurocholic and glycocholic acid into cholic acid and glycochenodeoxycholic and taurochenodeoxycholic acid into chenodeoxycholic acid. The importance of this activity lies in the fact that M-63 does not produce potentially hepatotoxic and carcinogenic secondary bile acids like deoxycholic and lithocholic acid after the complete biotransformation of bile salts [120], alleviating concerns regarding the safety of administering a secondary bile acid-producing bacterium.

Toxicological testing in animal models further underscores the safety of M-63. Both acute and chronic oral administration of M-63 to young rats did not result in mortality or toxic symptoms [121]. In the single dose oral toxicity test, five male and five female three-week-old Crj:CD (SD) IGS (SPF) rats were orally administered with a single dose of M-63 (3.2 × 10^11^ CFU/kg). No death and no abnormalities were observed. In the 90-day repeated-dose oral toxicity test using 7.5 × 10^10^ CFU/kg of M-63, no death and no abnormalities in body weight, food consumption, water consumption, urinalysis, hematology, blood biochemistry, organ weights, and histopathological parameters were observed. These findings demonstrate the absence of adverse effect by consumption of M-63. Additional in vitro and in vivo tests indicate that M-63 could adhere to mucin [123] but does not possess a mucin degradation ability [124], which is important as excessive degradation of this natural component of the gut barrier could lead to detrimental effects on the intestinal lining. Moreover, human clinical studies demonstrate that M-63 exerts positive impacts on gastrointestinal tolerance, regulation of gut microbiota, and contributing to a healthy state in infants and children. Altogether, M-63 stands as an efficient, safe, and practical probiotic strain for enhancing the nutrition and health of infants and children, as well as adults.

### 6.3. HMO Utilization Capacity of B. infantis M-63

One of the most unique features of the *B. infantis* species lie in its unique ability to consume all HMO structures in breast milk due to its unique genetic architecture [38,101,125,126,127]. Specifically, *B. infantis* contains several unique gene clusters (e.g., HMO cluster I or H1) that encode for intracellular GHs and ATP-binding cassette (ABC) transporters required for the capture, uptake, deconstruction, and metabolism of HMO [101,125,126,128,129] into acidic end products, mainly acetate and lactate, thereby lowering the colonic pH which then favors the colonization of other commensal bacteria [8].

Unlike most of the other infant-type HRB species, *B. infantis* efficiently consumes LNB and almost all types of HMOs, including fucosylated and sialylated molecules, intracellularly and such a consumption capability is highly and widely conserved in this subspecies [127,130,131,132]. Notably, *B. infantis* internalizes particularly intact small-mass HMOs, relying on the ABC transporters with a defined specificity for individual HMO families, and then degrade the oligosaccharides that have been taken up to monosaccharides intracellularly, using a repertoire of exo-acting GHs [125]. Nonetheless, the ability of *B. infantis* to utilize HMOs varies across different strains. Research has revealed that of the strains tested, only *B. infantis* ATCC 15697 and M-63 could ferment a range of HMOs including 3′-SL, 6′-SL, 2′-FL, and 3-FL [112]. More specifically, M-63 was particularly efficient in degrading almost 90% of 2′-FL, although it left most of the fucose in the supernatant [112]. Indeed, a recent study revealed that M-63 exhibited robust growth on 2′-FL, non-sialylated HMO, and sialylated HMO, demonstrating that M-63 degraded a majority of HMO structures after 10 hours of fermentation, with a preference for LNnT [113]. In contrast, the other strain, *B. infantis* R0033, showed only minimal growth on HMO and significant consumption of the fucosyllactoses after extended periods [113]. These findings exemplified the strain-specific capability of M-63 in HMO utilization. Further studies are needed to elucidate the mechanisms governing the catabolic process of HMO utilization in the strain M-63, which has a potential translational value for the rational selection of individual HMOs for synbiotic formulations.

Furthermore, a study that examined the growth of various bifidobacterial strains in human breast milk found that infant-type HRB strains, particularly M-63, exhibit a remarkable ability to thrive in breast milk [114]. The notable growth of M-63 can be largely attributed to its advanced capacity to metabolize HMOs, but its ability to withstand lysozyme also plays an essential role. Lysozymes are a common antibacterial agent in human breast milk, with levels as high as 400 µg/mL which are significantly higher than in cow’s milk [133]. This enzyme is also widespread in various tissues and bodily fluids including tears, saliva, sweat, and mucus [134]. As such, the tolerance of infant-type HRB strains like M-63 to lysozymes contributes to their ability to survive in the human body. In another study, the lysozyme tolerance in these HRB strains, including M-63, was found to stem from their ability to counteract the enzyme’s cationic properties linked with cytotoxicity [135]. This tolerance is key to how HRB strains like M-63 protect themselves against lysozyme, aiding their colonization in the human body. While further investigations are necessary to fully understand the exact mechanisms HRB strains use to tolerate lysozymes, the presence of lysozymes in human breast milk likely represents an important selection pressure favoring the colonization of infant-type HRB strains such as M-63.

Taken together, these illuminating findings underscore the significance of strain-level differences in the ability of *B. infantis* to consume HMOs and suggest that M-63, with its superior capacity to breakdown specific HMO compounds and tolerance to lysozymes, may have enhanced fitness and survival in the infant gut, thereby potentially offering better-associated health benefits throughout childhood (Figure 2). The intricate relationship between M-63, HMOs, and host health thus represents a symbiotic interaction which is crucial especially for infants, including premature neonates who rely on a balanced gut microbiota for optimal growth and development [8]. The unique abilities of M-63 thus make it an important subject of study in pediatric nutrition and a promising candidate for probiotic interventions.

### 6.4. Metabolite Production of B. infantis M-63

Through saccharolytic fermentation of carbohydrates in the gut (e.g., HMO metabolism), infant-type HRB species, including *B. infantis* produce metabolites including folate [115] and aromatic lactic acids such as indole-3-lactic acid (ILA), phenyllactate (PLA), and 4-hydroxypheyllactate (4-OH-PLA) [116,117] with physiological effects that may contribute to protection against infectious and immune-related diseases (Figure 2).

#### 6.4.1. Folate Production

Folate, also known as vitamin B9, is part of the water-soluble B-vitamin group. Folate is involved in critical cellular processes such as DNA synthesis, methylation, and repair, as well as the synthesis of nucleotides, certain vitamins, and amino acids, making it a critical nutrient during infancy and early childhood for proper development [136,137]. Cells that rapidly divide, such as blood and gut cells, especially need high folate levels to support their rapid growth [136]. Humans and other animals cannot synthesize folate naturally; it must be obtained through diet (like leafy greens, yeast, liver, and beans) or through beneficial gut bacteria capable of producing folate [138,139,140]. Inadequate folate levels are connected to numerous health issues, including neural tube defects during embryonic development, anemia, cancer risk, and cardiovascular illnesses [141,142,143]. It is notably crucial for population segments that are experiencing rapid cell growth, such as children, the elderly, and pregnant individuals, highlighting the importance of adequate folate intake for health maintenance.

Bifidobacteria are known for synthesizing a range of B vitamins, such as folate, biotin, thiamine, nicotinic acid, pyridoxine, riboflavin, and vitamin B12, although they have not been shown to produce vitamin K [140,144]. The study of bifidobacteria’s ability to both synthesize vitamins and release them outside the cell is still not thoroughly examined, with folate being the exception. Genomic analysis using the KEGG database has revealed that most lactobacilli species, except *Lactobacillus plantarum* strains, are missing the genes for pABA synthesis, as evidenced by the absence of the necessary enzymes for converting chorismate into pABA, suggesting they cannot produce folate without pABA and are dependent (auxotrophic) on external sources of folate [145]. On the other hand, several *Bifidobacterium* species, particularly the HRB species, are equipped to synthesize folate de novo [145,146]. While many bifidobacteria can produce the enzymes required for folate biosynthesis, not all have the complete set of enzymes.

Research has extensively evaluated the capability of various bifidobacteria species to synthesize folate and their potential role in augmenting the host’s folate levels to promote health [57,115,147]. Folate synthesis appears to be a trait confined to specific species or strains of bifidobacteria, with their capability to produce this vitamin being strongly correlated with their residential origins [115]. This correlation results in clear disparities in folate production among the high folate-producing HRB species and their non-HRB counterparts. For instance, while the majority of HRB strains including *B. infantis*, *B. bifidum*, *B. breve*, *B. longum*, and *B. adolescentis* can generate considerable amounts of folate with inter-strain variability, non-HRB strains tend to produce minimal amounts or are completely reliant on external sources for their folate supply [115,145,148]. Notably, infant-type HRB strains including *B. infantis* are noted as high folate producers when grown in media with low or absent folate content [145]. Considering that HRB strains with high folate-producing ability may serve as an endogenous source of folate that enhances the health of the host, it stands to reason that incorporating such strains, like M-63, could improve the intestinal homeostasis of this vitamin and provide health benefits during the critical stages of infancy and childhood when folate is a crucial nutrient. Additional research is required to clarify the specific advantages of folate-producing probiotic strains in fostering healthy development and possibly decreasing the likelihood of certain developmental problems.

#### 6.4.2. Aromatic Lactic Acids Production

The metabolites derived from bacterial tryptophan breakdown, such as aromatic lactic acids including indole-3-acetic acid (IAA), indole-3-aldehyde (IAld), indole-3-lactic acid (ILA), and indole-3-propionic acid (IPA), have been increasingly recognized for their critical role in maintaining both intestinal and systemic balance in the host [149,150]. These indoles act as aryl hydrocarbon receptor (AhR) ligands that may bolster gut barrier functions [150,151], modulate gut mucosal immunity [56,152,153,154,155], and potentially promote neurite outgrowth [156]. The production of these indole derivatives by HRB species is considered to be a key beneficial mechanism in the interaction between these microbes and the human host, suggesting a significant impact on health.

In breastfed infants, the gut is home to HMO-utilizing infant-type HRB species, such as *B. infantis,* which produce specific aromatic lactic acid metabolites, including ILA, phenyllactate (PLA), and 4-hydroxypheyllactate (4-OH-PLA) [116,117,157]. These metabolites, formed when bacteria ferment HMOs, and aromatic lactic acids like tryptophan, tyrosine, and phenylalanine from breast milk are known to interact with immune receptors, demonstrate immunomodulatory effects, and protect the gut lining [152,155,157,158]. Studies have shown that the production of these aromatic lactic acids are associated with the anti-inflammatory conditions in the gastrointestinal tracts of breastfed infants with abundant *Bifidobacterium* communities [56]. However, certain infants (such as those born prematurely or via C-section, or in high-antibiotic-use populations) may exhibit a delayed colonization or deficiency in *Bifidobacterium* species [159,160], which may lead to immune dysregulation [155] and allergy development [161]. Interestingly, the supplementation of HMO-utilizing *B. infantis* strains has been shown to regulate immune responses away from allergy-prone and autoimmune phenotypes [106,111,155]. Evidence from studies on both premature [102,103,104,105] and full-term infants [65,106] lacking *Bifidobacterium* suggests that introducing an HMO-utilizing infant-type HRB strain can suppress opportunistic pathogens and mitigate intestinal inflammation, underlining the critical role of the aromatic lactic acids produced by these bacteria play in infant immune homeostasis.

Notably, *B. infantis* M-63 has been found to possess the capability to produce a high level of ILA compared to other bifidobacterial strains [116,117]. This could be attributed to its superior capability in utilizing HMOs, as aforementioned. Other infant-type HRB strains of *B. longum*, *B. breve*, and *B. bifidum* also displayed a high capacity for ILA production [116]. Numerous studies have also demonstrated that infant-type HRB are particularly superior in producing ILA [162,163,164]. Additionally, in another study, M-63 was found to also capable of producing high levels of hydroxycarboxylic acid receptor 3 (HCA_3_) ligands including 4-OH-PLA, PLA, and ILA [117]. The HCA_3_ receptor, together with the other two receptors (HCA_1_ and HCA_2_) are recognized to play critical functions in the regulation of energy and immune system balance [165]. They are categorized as G-protein-coupled receptors based on their molecular structure [166]. While most mammals carry the HCA_1_ and HCA_2_ receptors, only humans and closely related hominids possess the HCA_3_ receptor [158]. HCA_3_, which is also called GPR109b, is present in various cell types including adipocytes, macrophages, neutrophils, and colonic epithelial cells [167]. The ability of M-63 to produce both AhR and HCA_3_ ligands implicates its potential mechanisms in interacting with the infant host where these ligands serve as signaling molecules to modulate intestinal and systemic immune responses, thereby contributing to infant and children’s health.

## 7. Clinical Effects of *B. infantis* M-63 on Infant and Children’s Health

Numerous clinical studies have been conducted to elucidate the safety and efficacy of M-63, involving a diverse group of more than 800 infants across various stages of development that included not only healthy full-term infants [65] but also those facing specific challenges such as extremely premature birth (gestation < 28 weeks) [102,103], low birth weight (<2000 g) [104], colic [107], allergies [106,110], and gastrointestinal surgeries [105] as well as more than 160 children aged 4–17 years old with allergies [111] or gastrointestinal issues [108,109]. Locations for these clinical studies have been widespread, encompassing Japan, France, Australia, Italy, and other countries. Consistently, these clinical studies have established that M-63 is safe and tolerated well with no adverse effects. Neither healthy nor ill infants exhibited adverse reactions after consuming this probiotic strain. Furthermore, M-63 exhibits beneficial effects in fostering the development of a healthy gut microbiota. When used in tandem with other probiotic strains, M-63 promotes bifidobacteria colonization, enhances gastrointestinal tolerance, and plays a role in supporting general health during infancy. Additionally, when paired with other infant-type HRB strains, M-63 shows substantial potential in reducing the symptoms associated with allergies and irritable bowel syndrome (IBS). Details of the clinical studies conducted with M-63 can be found in Table 1.

### 7.1. Fostering the Development of Healthy Infant Gut Microbiota

The initial acquisition of beneficial commensal bifidobacteria, particularly the infant-type HRB species such as *B. infantis*, is a crucial step for the healthy development of an infant [5]. This is evident for the probiotic strain M-63. In a double-blind, randomized, placebo-controlled study, supplementation with M-63 to healthy full-term infants within their first week until the age of 3 months significantly improved the formation of gut microbiota, the intestinal milieu, gastrointestinal functionality, and immune parameters [65]. The study involved 109 infants who were randomized into two groups, i.e., M-63 group or placebo group. Infants in each group were fed with M-63 (1 × 10^9^ CFU/1.0 g of sachet; *n* = 56) or placebo (sterilized dextrin only/1.0 g of sachet; *n* = 53) daily from within 7 days after birth to 3 months after birth. As a result, M-63 supplementation significantly increased the abundance of bifidobacteria in stools compared to the placebo, with positive associations noted concerning the frequency of breastfeeding [65]. This finding further supports the unique capability of M-63 in HMO utilization [112,113,114]. Additionally, M-63 supplementation at 1 month of age resulted in a reduction in stool pH and increased fecal acetic acid and secretory IgA (sIgA) levels as compared to the placebo group. Infants administered with M-63 also showed a reduction in the frequency of defecation and watery stools. No adverse effects were observed throughout the study period [65]. These results suggest that early supplementation with M-63 is safe, tolerated well, and contributes to the development of a bifidobacteria-dominant gut microbiota at a critical stage of development in full-term infants. M-63 also positively improves stool texture and enhances intestinal acetic acid and sIgA production, providing beneficial effects on gastrointestinal development, which is particularly essential in early infancy.

M-63 shows great promise in facilitating healthy gut microbiota development not only in healthy full-term infants but also in extremely premature infants [102,103] and low-birth-weight infants [104]. Premature infants are faced with considerable hurdles early in life, including being born prematurely, enduring extended hospital stays, potential immunodeficiency, antibiotic usage, and the postponement of enteral feeding [168]. Their underdeveloped gastrointestinal systems typically experience a delay in being colonized by beneficial commensal bacteria such as *Bifidobacterium* and *Bacteroides*, making them more prone to colonization by bacteria like *Enterobacteriaceae* and *Enterococcus* [169,170]. Moreover, the administration of antibiotics to these vulnerable infants can disrupt the natural colonization process of *Bifidobacterium* and lead to an increased presence of Proteobacteria, concurrently decreasing the overall diversity of the infant’s gut microbiota [171,172]. In light of these challenges, supplementation with M-63 appears to be a promising probiotic strategy that could mitigate the risks of gut dysbiosis, aid in the development of infant health, and promote growth.

A comparative, non-randomized controlled, prospective trial involving 44 low-birth-weight infants (body weight 1000–2000 g), who were ready for feeds within seven days of birth and were administered with either single strain of *B. breve* M-16V (5 × 10^8^ CFU/day; *n* = 15) or probiotics mixture containing three bifidobacterial strains, *B. infantis* M-63, *B. longum* BB536, and *B. breve* M-16V (5 × 10^8^ CFU/day of each strain; *n* = 13) or no administration as control group (*n* = 16) for six weeks, revealed a significant increase in the detection rates and cell numbers of bifidobacteria in the feces [104]. Notably, administration of a triple-strain probiotic mixture containing M-63 resulted in an earlier formation of a bifidobacteria-dominant microbiota and a significantly lower level of *Enterobacteriaceae* than the single-strain probiotic group. Over 85% of the low-birth-weight infants in the triple-strain group had detectable *B. infantis* M-63 and *B. breve* M-16V, while *B. longum* BB536 was found in 40% or fewer infants [104]. This study suggests that M-63 may act synergistically and cooperatively with other *Bifidobacterium* strains to facilitate the acquisition of bifidobacteria-dominant gut microbiota in premature infants, thereby promoting health.

Moreover, a double-blind, randomized, placebo-controlled trial involving 173 extremely preterm infants (gestation < 28 weeks) administered with either *B. breve* M-16V as single strain group or M-63, *B. longum* BB536, and *B. breve* M-16V as triple-strain group has also demonstrated similar findings where M-63, in combination with other *Bifidobacterium* strains, promoted bifidobacteria colonization and reduced dysbiotic condition in extremely preterm infants [103]. Both groups were administered with a total probiotic dose of 3 × 10^9^ CFU/day and extremely preterm infants from the placebo arm of a previous RCT were used as a reference group only for the microbiome analysis in this study. The primary outcome measured was the time to reach full feeding (TFF: 150 mL/kg/d), and secondary outcomes involved assessing fecal SCFAs and microbiota composition via 16S rRNA gene sequencing. The study found that both single-strain (*B. breve* M-16V) and triple-strain (containing M-63) probiotics were similar in timeline for achieving adequate feeding (median of 11 days) [103]. Significant increases in both propionic and butyric acid levels in the feces were observed from the initial week and 3 weeks after the single-strain and triple-strain probiotic supplementation. Although alpha diversity in fecal microbiota showed no discrepancy, beta diversity analyses indicated notable differences (*p* = 0.001) between the two probiotics and control groups, with a higher level of Actinobacteria (both *p* < 0.01) and lower levels of Gammaproteobacteria, Clostridia, and Negativicutes in both of the probiotics groups [103]. Conclusively, the study indicated that for extremely preterm infants, both single-strain and triple-strain (containing M-63) probiotics are beneficial in mitigating dysbiosis, characterized by an increase in beneficial bifidobacteria and a decrease in harmful Gammaproteobacteria.

In another comparative, non-randomized controlled, prospective trial examining the effects of the single-strain and triple-strain probiotic supplementation on the composition and volume of green gastric residuals, as well as feeding tolerance in extremely preterm infants (gestation < 28 weeks), the probiotics groups (both single- and triple-strain containing M-63) showed better outcomes in early enteral feedings [102]. The study involved 183 extremely preterm infants, with 154 receiving probiotics (single strain of *B. breve* M-16V: 75 and three-strain of M-63, *B. longum* BB536, and *B. breve* M-16V: 79) and 29 in the control group. Both groups were administered with a total probiotic dose of 3 × 10^9^ CFU/day. Extremely preterm infants given a placebo in a prior clinical trial served as the reference control for comparison. The study measured feeding outcomes such as the time to reach full feeding (TFF: 150 mL/kg/day) and the length of time on parenteral nutrition. As a result, the study found that extremely preterm infants who received both of the probiotics supplementation experienced a quicker median time to full enteral feeding (10 days vs. 14 days, *p* = 0.02) and shorter duration of parenteral nutrition (10 days vs. 16 days, *p* = 0.022) compared to the control group, indicating a potential effect of the probiotics, including M-63, in promoting faster attainment of feeding milestones in extremely preterm infants [102]. This could suggest that the supplementation of probiotics including M-63 may contribute to better outcomes in early enteral nutrition, as evidenced by the presence of bile acids in the green gastric residuals of these infants.

Additionally, M-63 also demonstrates great promise in attenuating gut dysbiosis and supporting healthy development in infants with challenging conditions such as those with congenital gastrointestinal surgical conditions (CGISC) [105]. These neonates often experience feed intolerance and healthcare-associated infections, and are at higher risks of developing gut dysbiosis due to frequent antibiotic use, delays in starting enteral feeds, dependency on parenteral nutrition, and hindered interactions with maternal skin and breast milk microbiota [173]. In this context, probiotic supplementation including M-63 has been shown to help reduce gut dysbiosis, thereby potentially improving the health outcomes for neonates with CGISC [105]. For instance, in a double-blind, randomized, placebo-controlled trial, 61 neonates diagnosed with congenital gastrointestinal surgical conditions (CGISC) were enrolled and randomized into two groups, with 30 in the probiotic group and 31 in the placebo group. The probiotic group were given a daily combination of triple-strain probiotics of M-63, *B. longum* BB536 at 1 × 10^9^ CFU/g for each strain. The control group was given maltodextrin as a placebo. This supplementation continued until hospital discharge. Fecal samples were collected at various intervals—prior to supplementation, after one week, after two weeks, and before discharge—and were analyzed utilizing 16S rRNA gene sequencing to evaluate fecal microbiota composition. The preliminary outcomes of this study highlighted that after two weeks of receiving a three-strain probiotics supplementation containing M-63, neonates with CGISC exhibited a lower prevalence of potentially pathogenic bacteria in their stools, an increased abundance of *Bifidobacterium*, and heightened levels of SCFAs [105]. While the findings are promising, larger studies with defined clinical outcomes and extended follow-up periods are necessary.

In another study, M-63 was found to colonize the gastrointestinal tract and persisted in the gut microbiota of infants with cow’s milk allergy (CMA) for 60 days [106]. The results indicate that M-63 has the capability to traverse the gastrointestinal tract and remain for an extended duration within the intestines of children who have a cow’s milk allergy. This persistence of M-63 is linked to alterations in the gut microbiota, notably increased levels of *Akkermansia* spp. and *Ruminococcus* spp. This study was a prospective, multicenter comparison of gut microbiota in infants with CMA, infants with cow’s milk sensitization, and healthy infants that examined the effects of triple-strain probiotic mixtures containing M-63 on regulating gut microbiota and maintaining probiotics persistence in CMA patients. The study characterized the gut microbiota of infants with CMA by an imbalance of beneficial bacteria and a dominance of certain pathogenic bacteria such as *Haemophilus*, *Klebsiella*, and *Prevotella* [106]. Globally, the incidence of IgE-mediated CMA in infants and young children is estimated to be around 2–3% [174]. Children diagnosed with CMA during their first year have an elevated risk of developing other atopic diseases later on [175,176]. There are several epidemiological indicators that suggest a potential connection between CMA and changes in gut microbiota [42]. Early microbial exposure tends to shift the immune balance towards a Th1 phenotype, whereas unsuccessful normal gut bacterial colonization leans towards a Th2 response, which can perpetuate allergic conditions [177]. Given this context, probiotics supplementation such as M-63 serves as a promising means to prevent food allergies by potentially stimulating the immune responses governed by Th1. Over the course of the study, 124 infants aged 10–15 months were screened, culminating in the enrollment of 40 infants as follows: 14 with CMA, 12 with CMS, and 14 healthy controls. Infants in the CMA group were given a triple-strain probiotics mixture containing M-63, *B. longum* BB536, and *B. breve* M-63, with a daily dose of 3.5 × 10^9^ CFU of each strain twice daily for a month. The CMS group had no dietary restrictions and received diets with non-functional food formula substitutes, while the healthy group was provided regular formula. Notably, M-63 successfully colonized the gastrointestinal tract and was detectable in the gut microbiota of the infants in the CMA group for up to 60 days, highlighting its beneficial effect on promoting healthy gut microbiota development and immune tolerance [106].

Collectively, these data have exemplified that M-63 is potentially beneficial at promoting early colonization of bifidobacteria and may therefore support healthy growth in not only healthy full-term infants but also premature infants and infants with challenging conditions.

### 7.2. Enhancing Gastrointestinal Tolerance in Infants

Colic in infants, marked by excessive crying and abdominal distress without an established cause, is common and potentially linked to differences in the gut microbiome [178]. Infants with colic have been found to have fewer beneficial gut bacteria like *Bifidobacterium*, *Lactobacillus*, and *Faecalibacterium prausnitzii* [179]. This connection was explored in a double-blind, randomized, multicenter controlled trial examining the impact of probiotic supplementation with M-63 and *Lactobacillus rhamnosus* LCS-742 in an infant formula enriched with α-lactalbumin-enriched on colic, the nutritional adequacy, and gastrointestinal tolerance [107]. The study included 66 colicky infants aged from 3 weeks to 3 months, who received either an experimental formula with added α-lactalbumin, M-63 (10^7^ CFU/g), and *L. rhamnosus* LCS-742 (10^7^ CFU/g), or a control formula for a month. The EF was lower in protein and lactose, while the CF was higher in these components and lacked the additional α-lactalbumin and probiotics. Notably, infants receiving the EF showed improved gastrointestinal tolerance with fewer feeding-related gastrointestinal side effects, although there was no significant change in crying duration between the two groups [107]. Taken together, this study suggests that formula enriched with α-lactalbumin and probiotics such as M-63 not only supports the appropriate height and weight increase but also improves gastrointestinal tolerance in infants suffering from colic.

In a similar study, in which 97 full-term infants were administered a formula augmented with M-63 and *L. rhamnosus* LCS-742 (140 million CFU per 100 mL) for six months, the results after one month showed reduced crying or restlessness and a quieter behavior (*p* = 0.03), with a decreased incidence of atopic dermatitis at six months (*p* < 0.05) [110]. Together, these findings suggest the beneficial effects of the probiotic mixture containing M-63 in treating colic and potentially acting as a protective agent against atopic dermatitis.

### 7.3. Supporting Children’s Health

Several interventional studies suggest that M-63 in a probiotic mixture of *B. longum* BB536 and *B. breve* M-16V could alleviate allergic disorders, including allergic rhinitis and intermittent asthma [111], as well as improve IBS [108] and functional constipation conditions [109]. In a placebo-controlled, double-blinded and randomized trial involving 40 Italian children (mean age 9 ± 2.2 years) treated with a probiotic mixture containing M-63 (1 × 10^9^ CFU/day), *B. longum* BB536 (3 × 10^9^ CFU/day), and *B. breve* M-16V (1 × 10^9^ CFU/day) for four weeks, administration of the probiotic mixture containing M-63 protected the children against pollen-induced IgE-mediated allergic rhinitis and intermittent asthma and improved their quality of life, while these parameters were worsened in the placebo group. The findings highlight that M-63 could also dampen allergic disorders and improve children’s quality of life when combined with other *Bifidobacterium* strains, which is in line with the guidelines from the World Allergy Organization to use probiotics as a preventive approach for high-risk infants and children [180].

On the other hand, the same probiotic mixture containing M-63 also shows great promise in alleviating gastrointestinal disorders such as chronic functional constipation [109] and IBS [108]. Nearly a third of children with constipation continue experiencing symptoms into adolescence [181]. As a result, there is growing interest in using probiotics as an adjunct therapy alongside polyethylene glycol (PEG) to potentially avoid any PEG-related imbalance in gut bacteria [182]. In a randomized controlled study, the effect of the probiotic mixture containing M-63 (1 × 10^9^ CFU/day), *B. longum* BB536 (3 × 10^9^ CFU/day), and *B. breve* M-16V (1 × 10^9^ CFU/day) on children with functional constipation was evaluated. Fifty-five children aged 4-12 years received either just PEG or PEG combined with the probiotic mixture containing M-63. After eight weeks, improvements were similar in both groups, but a slight trend suggested greater long-term remission rates in the probiotics group [109]. The study concluded that while both treatments are safe and effective in the short term, further research is needed to determine if probiotics can improve long-term outcomes.

In another controlled trial with a crossover design, a group of children between the ages of 8 and 16 years suffering from IBS (48 participants) and functional dyspepsia (25 participants) were given the same probiotics mixture containing M-63 (1 × 10^9^ CFU/day), *B. longum* BB536 (3 × 10^9^ CFU/day), and *B. breve* M-16V (1 × 10^9^ CFU/day) or placebo over the course of six weeks [108]. The results indicated that the probiotic mixture containing M-63 were effective in reducing abdominal pain for those with IBS but did not provide the same benefit for children with functional dyspepsia. Moreover, a significantly larger number of children with IBS reported an enhanced quality of life after taking the probiotics compared to those who received a placebo, with 48% noting improvement versus 17% [108]. Taken together, these clinical findings support the notion that supplementation with the probiotic mixture containing M-63 can be a potential approach to improve gastrointestinal and allergic conditions, although larger clinical trials are needed for definite confirmation, especially on the single-strain effect. These clinical studies serve as a basis to incorporate M-63 in children formula and supplement products as a means to promote children’s health and consequently protect high-risk children from different kinds of complications.

## 8. Application of *B. infantis* M-63

M-63 was deemed “Generally Recognized as Safe” (GRAS) by the United States Food and Drug Administration (FDA) in 2022 for use in infant formula and general foods (GRN No., 1003) [120]. Moreover, the strain is approved for food use across multiple countries, including China, the European Union, Canada, and Australia, demonstrating its global safety and application profile. For instance, the consistent use of M-63 in infant formula milk powder since 2006 in Spain, France, and Indonesia, as well as in infant nutritional supplements in the United States and Italy since 2008 and 2009, underlines its established history of consumption without adverse feedback.

## 9. Conclusions

In conclusion, the unique physiological features of *B*. *infantis* M-63 underscore its superiority as an infant-type HRB strain (Figure 3). M-63 demonstrates the exceptional capability of utilizing HMOs, the complex carbohydrates found in human breast milk, setting it apart from other probiotic strains. Furthermore, M-63 can also produce beneficial metabolites such as folate and aromatic lactic acids like ILA, contributing to the overall well-being and immune development in infants and children. In clinical settings, the supplementation of M-63 has demonstrated remarkable benefits in promoting infant and children’s health. The findings from various studies have consistently shown its effectiveness as a single strain or in combination with other probiotic strains in promoting a bifidobacteria-dominant gut microbiota, improving gastrointestinal tolerance, alleviating allergic symptoms.

The apparent decline of the *B. infantis* species in industrialized nations is a concerning trend, considering its significant impact on infant and children’s development. Given the compelling evidence of the clinical benefits of M-63, which is also an efficient HMO utilizer, the incorporation of M-63 in combination with HMOs in infant and children’s formula is much anticipated. Supplementation with M-63 and HMOs holds the potential to promote healthy growth and development throughout childhood. Further large-scale clinical trials would help to confirm the specific effects of M-63 in combination with specific HMOs in emulating the health benefits observed in vaginally delivered and breastfed healthy infants and their potential as a preventive approach in promoting children’s health and protecting them from various complications.

## Figures and Tables

**Figure 1 microorganisms-12-01014-f001:**
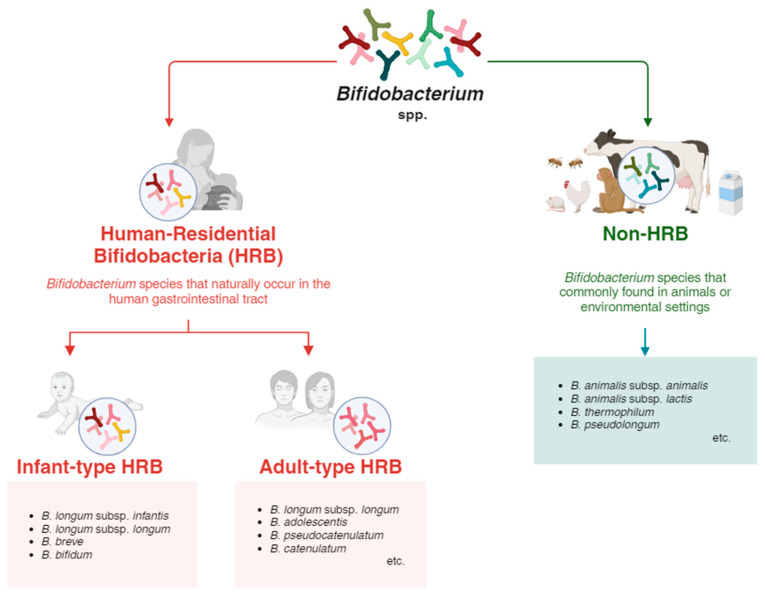
Distinctive differences in the ecological distribution of bifidobacteria. In the human host, the *Bifidobacterium* species that naturally occur in the human gastrointestinal tract are referred to as the human-residential bifidoabcteria (HRB). Within the HRB group, the bifidobacterial species that are prevalent in the infant gut, including *Bifidobacterium longum* subsp. *infantis*, *B. breve*, *B. bifidum*, and *B. longum* subsp. *longum*, are sub-categorized as infant-type HRB. Conversely, other bifidobacterial species that are more typically found in the intestines of adults, including *B. longum* subsp. *longum*, *B. adolescentis*, *B. catenulatum*, *B. pseudocatenulatum*, etc., are grouped as adult-type HRB. In contrast, bifidobacterial species that are commonly found in animals or environmental settings, including *B. animalis* subsp. *animalis*, *B. animalis* subsp. *lactis*, *B. thermophilum*, *B. pseudologum*, etc., are referred to as non-HRB species. The species of HRB and non-HRB display differences in their ecological adaptation.

**Figure 2 microorganisms-12-01014-f002:**
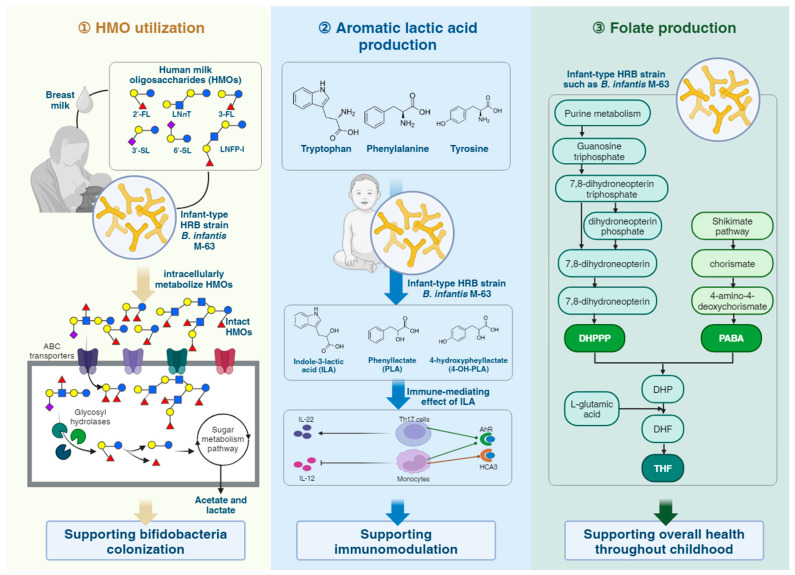
The physiological properties of *Bifidobacterium longum* subsp. *infantis* M-63 on HMO utilization and metabolite production such as aromatic lactic acids and folate. (**1**) M-63 demonstrates superiority in utilizing HMO compounds internally that may have enhanced its fitness and survival in the infant gut, contributing to its beneficial effect on bifidobacteria colonization. (**2**) M-63 is also capable of producing aromatic lactic acids including indole-3-lactic acid (ILA), phenyllactate (PLA), and 4-hydroxyphenyllactate (4-OH-PLA) that are associated with immune-modulating effects. (**3**) M-63 also potentially produces folate, a critical cofactor, and thereby potentially offering better-associated health benefits throughout childhood.

**Figure 3 microorganisms-12-01014-f003:**
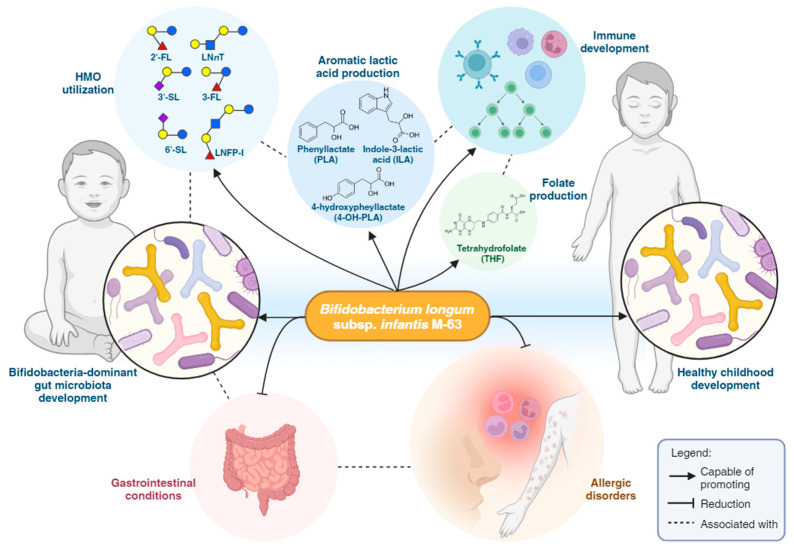
The multifaceted physiological effects of *Bifidobacterium longum* subsp. *infantis* M-63 on human milk oligosaccharide (HMO) utilization and metabolite production and its clinical benefits throughout childhood. The specialized enzymatic machinery of M-63 enables it to efficiently metabolize HMOs, promoting the development of a bifidobacteria-dominant gut microbiota and immune development in early life. The strain also produces key metabolites such as folate, which is essential for DNA synthesis, cell growth, and neurological development as well as aromatic lactic acid including phenyllactate, indole-3-lactic acid (ILA), and 4-hydroxyphenyllactate (4-OH-PLA), which are essential for immune function. Clinically, supplementation with M-63 has been shown to lead to a bifidobacteria-dominant gut microbiota, reduced gastrointestinal discomforts, reduced allergic symptoms, and healthy childhood development. Studies have demonstrated its effectiveness both as a single probiotic and in combination with other strains to support overall health status of a child.

**Table 1 microorganisms-12-01014-t001:** Summary of the clinical studies of *Bifidobacterium infantis* M-63 on infant and children’s health.

Reference	Study Design	Study Characteristics
*Effect of M-63 on the development of healthy infant gut microbiota*		
Hiraku et al., 2023 [65]	Randomized, double-blinded, placebo-controlled	Study population: Healthy full-term infants
		Country: Japan
		Sample size: *n* = 109 (Probiotics: 56; Placebo: 53)
		Intervention and dose: M-63 (1 × 10^9^ CFU/day/1.0 g of sachet) or placebo (sterilized dextrin only/1.0 g of sachet)
		Duration of supplementation: Supplementation started in the first week following birth until the age of 3 months.
		Main outcomes: (1) Significant increase in the abundance of bifidobacteria in stools after M-63 supplementation. (2) A reduction in stool pH, increased fecal acetic acid and secretory IgA (sIgA) levels as compared to the placebo group. (3) A reduction in the frequency of defecation and watery stools after M-63 supplementation. (4) No adverse effects were observed in the M-63 group throughout the study period.
Ishizeki et al., 2023 [104]	Comparative, non-randomized controlled, prospective	Study population: Low birth weight infants (birth weight: 1000–2000 g)
		Country: Japan
		Sample size: *n* = 44 (Single-strain probiotics: 15; Triple-strain probiotics: 13, Control: 16)
		Intervention and dose: (1) Control group: No probiotics (2) Single strain *B. breve* M-16V group: 5 × 10^8^ CFU/day in 1.5 mL sterile water (3) Three-strain probiotics mixture group: *B. infantis* M-63, *B. longum* BB536, and *B. breve* M-16V; 5 × 10^8^ CFU/day of each strain in 1.5 mL sterile water
		Duration of supplementation: Six weeks.
		Main outcomes: (1) Earlier formation of a bifidobacteria-dominant microbiota and a significant lower level of Enterobacteriaceae in triple-strain probiotics group. (2) Over 85% of the low-birth-weight infants in the triple-strain group had detectable *B. infantis* M-63 and *B. breve* M-16V, while *B. longum* BB536 was found in 40% or fewer infants.
Athalye-Jape et al., 2022 [103]	Randomized, double-blinded, placebo-controlled	Study population: Extremely preterm neonates (gestation < 28 weeks)
		Country: Australia
		Sample size: *n* = 173 (Single-strain probiotics: 86; Triple-strain probiotics: 87)
		Intervention and dose: (1) Control group: No probiotics (2) Single strain *B. breve* M-16V group: 3 × 10^9^ CFU/day (3) Three-strain probiotics mixture group: *B. infantis* M-63, *B. longum* BB536, and *B. breve* M-16V; total 3 × 10^9^ CFU/day
		Duration of supplementation: Probiotics supplementation commenced with feeds and continued until 37 weeks of corrected gestational age.
		Main outcomes: (1) Similar timeline in achieving adequate feeding in both single-strain and triple-strain probiotics groups (median of 11 days). (2) Significant increase in the abundance of Actinobacteria and lower levels of Gammaproteobacteria, Clostridia, and Negativicutes in both of the probiotics groups.
Athalye-Jape et al., 2020 [102]	Comparative, non-randomized controlled, prospective	Study population: Extremely preterm neonates (gestation < 28 weeks)
		Country: Australia
		Sample size: *n* = 183 (Single-strain probiotics: 75; Triple-strain probiotics: 79, Control: 29)
		Intervention and dose: (1) Control group: No probiotics (2) Single strain *B. breve* M-16V group: 3 × 10^9^ CFU/day (3) Three-strain probiotics mixture group: *B. infantis* M-63, *B. longum* BB536, and *B. breve* M-16V; total 3 × 10^9^ CFU/day
		Duration of supplementation: Probiotics supplementation commenced with feeds and continued until time to reach full feeding (TFF: 150 mL/kg/day).
		Main outcomes: (1) Significant shorter median time to full enteral feeding and duration of parental nutrition in both probiotic groups. (2) Bile acid content was higher but statistically insignificant in dark vs. pale green gastrointestinal residuals in both probiotic groups.
Rao et al., 2022 [105]	Randomized, double-blinded, placebo-controlled	Study population: Neonates (≥35 weeks) with congenital gastrointestinal surgical conditions (CGISC)
		Country: Australia
		Sample size: *n* = 61 (Probiotics: 30; Placebo: 31)
		Intervention and dose: Triple-strain probiotics containing M-63, *B. longum* BB536 and *B. breve* M-16V (each strain at 1 × 10^9^ CFU/day/g of sachet) or placebo (maltodextrin)
		Duration of supplementation: Supplementation continued until hospital discharge.
		Main outcomes: (1) Significant lower abundance of potentially pathogenic bacteria in probiotics group. (2) Significant increase in the abundance of bifidobacteria and levels of short-chain fatty acids in probiotics group.
Mennini et al., 2021 [106]	Prospective, multicenter comparison	Study population: Infant aged 10–15 months old with cow’s milk allergy
		Country: Italy
		Sample size: *n* = 40 Cow’s milk allergic group with probiotics: 14 Cow’s milk sensitized group: 12 Healthy group: 14
		Intervention and dose: Cow’s milk allegic infant were given a mixture of triple-strains probiotics containing M-63, *B. longum* BB536 and *B. breve* M-16V (each strain 3.5 × 10^9^ CFU/dose, twice daily)
		Duration of supplementation: 30 days
		Main outcomes: (1) M-63 colonized and persisted in the gastrointestinal tract of infants with cow’s milk allergy for 60 days. (2) The persistence of M-63 is linked to an increased level of *Akkermansia* spp. and *Ruminococcus* spp.
*Effects of M-63 on gastrointestinal tolerance in infants*		
Dupont et al., 2010 [107]	Randomized, double-blinded, multicenter controlled	Study population: Colicky infants aged 3 weeks to 3 months
		Country: France
		Sample size: *n* = 66 (Probiotics: 33; Placebo: 33)
		Intervention and dose: An experimental formula (EF) with added α-lactalbumin, M-63 (10^7^ CFU/g), and *L. rhamnosus* LCS-742 (10^7^ CFU/g), or a control formula
		Duration of supplementation: One month
		Main outcomes: (1) Significant improvement in gastrointestinal tolerance in infants receiving the EF with fewer feeding-related gastrointestinal side effects (2) No significant change in crying duration between the two groups. (3) Supplementation of EF supported appropriate height and weight increase in colicky infants.
Roze et al., 2012 [110]	Randomized, double-blinded, multicenter controlled	Study population: Non-breastfed full-term infants
		Country: France
		Sample size: *n* = 97 (Probiotics: 48; Placebo: 49)
		Intervention and dose: An experimental formula (EF) with added α-lactalbumin, M-63 and *L. rhamnosus* LCS-742 140 million CFU per 100 mL or a control formula
		Duration of supplementation: 6 months
		Main outcomes: (1) Significant reduction in crying or restlessness and more quiet behavior in the EF group. (2) Significant reduction in incidence of atopic dermatitis in the EF group.
*Effects of M-63 on children’s health*		
Miraglia Del Giudice et al., 2017 [111]	Randomized, double-blinded, placebo-controlled	Study population: Children aged 4–17 years
		Country: Italy
		Sample size: *n* = 40 (Probiotics: 20; Placebo: 20)
		Intervention and dose: Triple-strain probiotics mixture containing M-63 (1 × 10^9^ CFU/day), *B. longum* BB536 (3 × 10^9^ CFU/day), and *B. breve* M-16V (1 × 10^9^ CFU/day)
		Duration of supplementation: 4 weeks
		Main outcomes: (1) Significant reduction in prevalence of pollen-induced IgE-mediated allergic rhinitis and intermittent asthma in the probiotics group (2) Significant improvement of quality of life in the probiotics group
Russo et al., 2017 [109]	Randomized controlled	Study population: Children aged 4–12 years with functional constipation
		Country: Italy
		Sample size: *n* = 55 (Probiotics: 27; Control: 28)
		Intervention and dose: either just PEG or PEG combined with the probiotic mixture containing M-63 (1 × 10^9^ CFU/day), *B. longum* BB536 (3 × 10^9^ CFU/day), and *B. breve* M-16V (1 × 10^9^ CFU/day)
		Duration of supplementation: 8 weeks.
		Main outcomes: (1) Similar improvements similar in both groups at week 8. (2) A slight trend of greater long-term remission rates in the probiotics group.
Giannetti et al., 2017 [108]	Crossover, placebo-controlled	Study population: Children aged 8–16 years
		Country: Italy
		Sample size: *n* = IBS (48 participants) and functional dyspepsia (25 participants)
		Intervention and dose: A probiotics mixture containing M-63 (1 × 10^9^ CFU/day), *B. longum* BB536 (3 × 10^9^ CFU/day), and *B. breve* M-16V (1 × 10^9^ CFU/day) or a placebo
		Duration of supplementation: 6 weeks
		Main outcomes: (1) Significant reduction in abdominal pain for those with IBS and received the probiotics but not functional dyspepsia. (2) Significantly larger number of children with IBS reported an enhanced quality of life after taking the probiotics compared to those who received a placebo (48% vs. 17%).

## Data Availability

Not applicable.

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
