# Peer review of "Probiotics in the New Era of Human Milk Oligosaccharides (HMOs): HMO Utilization and Beneficial Effects of *Bifidobacterium longum* subsp. *infantis* M-63 on Infant Health"

_microorganisms, 2024, doi:10.3390/microorganisms12051014_

Round 1

Reviewer 1 Report

Comments and Suggestions for Authors

Title: " Probiotics in the new era of human milk oligosaccharides 2 (HMOs): HMO utilization and beneficial effects of Bifidobacte- 3 rium longum subsp. infantis M-63 on infant health

 This review focuses on the emerging role of probiotics in conjunction with human milk oligosaccharides (HMOs) in infant health. It specifically examines the utilization of HMOs and the beneficial effects of Bifidobacterium longum subsp. infantis M-63 on infants.

Minor suggestions

Graphical Abstract: The figure in the abstract does not match the content.

To address the discrepancy between the figure and the abstract, you can consider moving the figure to the end of the review as a summary of the references mentioned in the manuscript. This way, the figure will align with the content and provide a visual representation of the referenced studies.

 Line 58: delete [8] since it is named again at the end of the paragraph.

Author Response

Graphical Abstract: The figure in the abstract does not match the content.

To address the discrepancy between the figure and the abstract, you can consider moving the figure to the end of the review as a summary of the references mentioned in the manuscript. This way, the figure will align with the content and provide a visual representation of the referenced studies.
⇒ Thank you for pointing this out. We agreed with the suggestion and have moved the figure to the end of the review as a summary. We included a footnote as follows.

Figure 3. The multifaceted physiological effects of Bifidobacterium longum subsp. infantis M-63 on human milk oligosaccharide (HMO) utilization and metabolite production and its clinical benefits throughout childhood. The specialized enzymatic machinery of M-63 enables it to efficiently metabolize HMOs, promoting the development of a bifidobacteria-dominant gut microbiota and immune development in early life. The strain also produces key metabolites such as folate, which is essential for DNA synthesis, cell growth, and neurological development as well as aromatic lactic acid including phenyllactate, indole-3-lactic acid (ILA), and 4-hydroxyphenyllactate (4-OH-PLA), which are essential for immune function. Clinically, supplementation with M-63 has been shown to lead to a bifidobacteria-dominant gut microbiota, reduced gastrointestinal discomforts, reduced allergic symptoms, and healthy childhood development. Studies have demonstrated its effectiveness both as a single probiotic and in combination with other strains to support the overall health status of a child.

 Line 58: delete [8] since it is named again at the end of the paragraph.
⇒ This was deleted. 

Reviewer 2 Report

Comments and Suggestions for Authors

This article examines the decline of beneficial gut bacteria, specifically Bifidobacterium longum subsp. infantis (B. infantis), in infants from industrialized nations and emphasizes the importance of its restoration. It discusses the potential of B. infantis strains, which utilize Human Milk Oligosaccharides (HMOs) and positively impact infant health. Additionally, the article evaluates the role of HMOs in infant formulas and proposes combining B. infantis M-63 with HMOs to enhance health outcomes in infants and children. The paper is well-written, featuring meticulously organized materials and methods, appropriate statistical analyses, and a comprehensive discussion. The up-to-date references in the bibliography enhance the paper's credibility.
I recommend accepting and publishing it.

Author Response

Thank you for taking the time to review the manuscript and providing your valuable comments. 
